# MODELING ABSTRACT STYLE PROMPTS FOR TEXT-TO-SPEECH MODELS

## ABSTRACT

A recent trend in text-to-speech synthesis (TTS) is to construct models capable of generating naturalistic speech that adheres to a textual style prompt describing the speaker's voice and speaking style. In this paper, we propose a crisper definition of style-prompted TTS by categorizing style tags by how they can be collected (*automatic* tags obtainable using signal processing tools e.g. low-pitched and slow; *demographic* tags obtainable using speaker demographics e.g. male and American accent; and *abstract* tags which need human-annotations e.g. authoritative and awed) and what they represent (*intrinsic* tags inherent to speaker identity e.g. gender, average pitch, texture; and *situational* tags specific to utterance-level speaking styles e.g. emotion). Compared to previous work, we expand the space of style prompts substantially by covering 47 abstract tags, 10 demographic tags and 6 automatic tags. For abstract intrinsic tags, we annotate a subset of speakers from the VoxCeleb (Nagrani et al., 2020) dataset. For abstract situational tags, we leverage existing speaking-style-based datasets Expresso (Nguyen et al., 2023) and EARS (Richter et al., 2024). We train a style-prompted TTS model based on Parler-TTS (Lyth & King, 2024; Lacombe et al., 2024b) using these datasets and find that our model outperforms baselines on speech-style consistency metrics. Our collected dataset and model will be open-sourced.

## 1 INTRODUCTION

Text-to-speech systems that are controllable by natural language text style prompts a.k.a. style-prompted TTS systems e.g. (Guo et al., 2022; Ji et al., 2024; Leng et al., 2023; Vyas et al., 2023; Lacombe et al., 2024b; Jin et al., 2024) have been gaining prominence in the past few years. Rather than providing control via a few seconds of reference speech (Peng et al., 2024; Wang et al., 2023) exhibiting the desired style, these models allow users to do so via natural language instead, which provides a more explicit, intuitive, and privacy-preserving control medium. Training these models requires a dataset which has speech utterances annotated with natural language style prompts.

When humans describe speech speaking styles in natural language, they do so with a rich and diverse vocabulary spanning a wide range of style tags covering aspects like pitch, texture, emotion and rhythm and more. We propose a crisper definition of style-prompted TTS that rigorously categorizes style tags along two axes: the mechanism by which one can obtain tag annotations (*automatic*, *demographic* and *abstract* tags) and what speech aspects the tag represents (*intrinsic* tags and *situational* tags). Based on an extensive survey of previous style-prompted TTS work (Section 2), we find that existing work offers natural-language control over some, but not all of these categories, often overlooking the importance of covering more abstract speech style tags that cannot be automatically extracted. More importantly, the only current open-source model, Parler-TTS (Lyth & King, 2024; Lacombe et al., 2024b) only supports automatic tags, which motivates the need for an open-source model that can support all categories.

We create a list of 63 style tags consisting of 26 abstract intrinsic, 21 abstract situational, 10 demographic and 6 automatic tags covering a wide variety of speech styles. As described in Section 3, to support all these tags, we collect abstract intrinsic human annotations for a subset of the Vox-Celeb (Nagrani et al., 2020) dataset, creating StyledVoxCeleb. While not originally proposed for style-prompted TTS, Expresso (Nguyen et al., 2023) and EARS (Richter et al., 2024) cover a rich variety of abstract situational speaking styles and hence we reuse them for TTS. We finetune Parler-

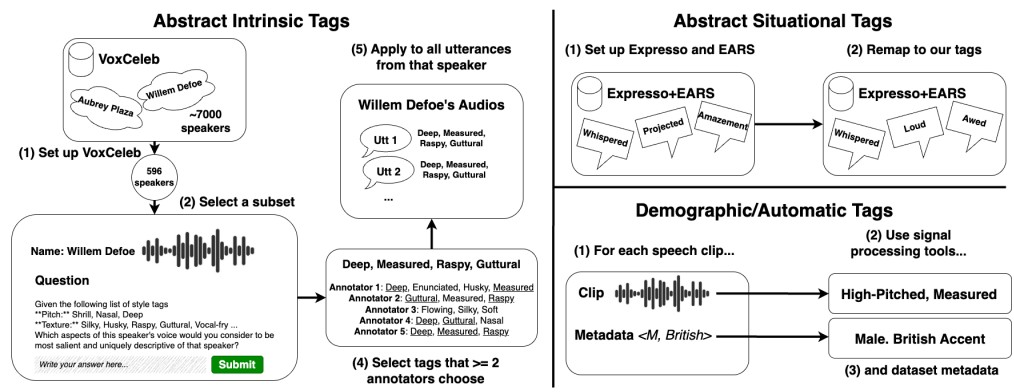

Figure 1: An overview of our data collection procedure.

TTS (Lacombe et al., 2024b; Lyth & King, 2024) on these datasets and find (Section 5) that our model outperforms competitive baselines on speech-style consistency metrics.

In summary, our contributions are:

- We provide a crisper categorization of style tags and perform an extensive survey of prior work based on this categorization.
- We create StyledVoxCeleb, a subset of the VoxCeleb (Nagrani et al., 2020) dataset annotated with abstract intrinsic style tags, reuse Expresso and EARS for TTS, and train style-prompted TTS model that cover all categories of speech style tags.
- We demonstrate that training on our dataset pool results in improved performance on speech-style consistency metrics, obtaining $+0.1$ in consistency MOS and $+0.06$ tag recall (metrics introduced in Section 4) as compared to the next best baseline.

We will open-source our model and collected data upon publication.

## 2 BACKGROUND AND MOTIVATION

We can describe speech styles with a rich and diverse vocabulary, capturing aspects such as pitch, emotion, rhythm, speaking rate and more. We draw a distinction between *intrinsic* tags that are tied to a speaker's identity and persist across all utterances belonging to that speaker (e.g. average pitch and vocal texture) and *situational* tags that describe the speaking style (e.g. emotion) of individual utterances. [1] This distinction is important for deciding a data collection strategy; while intrinsic tags can be annotated on a per-speaker basis, situational tags must be annotated on a per-utterance basis, which is significantly more expensive. Furthermore, depending on how the tags can be collected, we distinguish between *abstract* tags that are complex and require human annotations (e.g. clarity, texture, emotion) , *demographic* tags obtainable from speaker demographics (e.g. gender and accent) and *automatic* tags [2] that are obtainable via signal processing tools (e.g. pitch/F0, energy, speaking rate). Abstract and automatic tags can be both intrinsic and situational, while demographic tags are always intrinsic since they pertain to speaker demographics. In the rest of this paper, we classify tags into four categories: *abstract intrinsic*, *abstract situational*, *demographic* and *automatic* tags (combining intrinsic and situational automatic tags).

Style-prompted TTS should support a diverse space of speech style prompts, covering many tags across all categories. We perform a comparison of prior work summarized in Table 1 and make several observations. First, we notice that none of the previous datasets apart from AudioBox and

---

[1] Some tag categories can be both intrinsic and situational and need to be handled carefully; see Appendix A.

[2] While there exist automatic emotion classifiers (Ma et al., 2023) for a subset of emotions used by prior work e.g. (Jin et al., 2024), we found that their quality on our datasets is unsatisfactory. In this work, we still consider them to be abstract tags.

| Dataset | Abst | | Dem | Auto | # Abst | Open-Sourced | |
|---|---|---|---|---|---|---|---|
| | Intr | Sit | | | | Data | Model |
| PromptSpeech (Guo et al., 2022) | ✗ | ✓ | ✓ | ✓ | 4 | ✓ | ✗ |
| NLSpeech (Yang et al., 2023) | ✗ | ✓ | ✓ | ✓ | ? | ✗ | ✗ |
| PromptStyle (Liu et al., 2023) | ✗ | ✓ | ✓ | ✓ | ? | ✗ | ✗ |
| TextrolSpeech (Ji et al., 2024) | ✗ | ✓ | ✓ | ✓ | 8 | ✓ | ✗ |
| Coco-Nut (Watanabe et al., 2023) | ✓ | ✓ | ✓ | ✓ | ? | ✓ | ✗ |
| PromptTTS2 (Leng et al., 2023) | ✗ | ✗ | ✓ | ✓ | 0 | ✗ | ✗ |
| MEAD-TTS (Guan et al., 2024) | ✗ | ✓ | ✓ | ✓ | 8 | ✓ | ✗ |
| AudioBox (Vyas et al., 2023) | ✓ | ✓ | ✓ | ✓ | ? | ✗ | ✗ |
| ParlerTTS (Lacombe et al., 2024b) | ✗ | ✗ | ✓ | ✓ | 0 | ✓ | ✓ |
| LibriTTS-P (Kawamura et al., 2024) | ✓ | ✗ | ✓ | ✓ | 44 | ✓ | ✗ |
| SpeechCraft (Jin et al., 2024) | ✗ | ✓ | ✓ | ✓ | 7 | ✓ | ✗ |
| Ours | ✓ | ✓ | ✓ | ✓ | 47 | ✓ | ✓ |

Table 1: A comparison of existing style-prompted TTS papers. The # Abst column denotes the number of abstract tags in each dataset. We denote with the ? symbol those datasets whose abstract tag count is unknown. To the best of our knowledge, only AudioBox and Coco-Nut cover all three abstract tag categories. Of these, the AudioBox dataset is closed-source and Coco-Nut is only 8 hours long, making both unusable for training a TTS model. While LibriTTS-P (Kawamura et al., 2024) has nearly as many tags as ours, it does not cover abstract situational tags.

Coco-Nut cover all categories. Both are unusable for training TTS models, since AudioBox is closed-source and Coco-Nut is only 8 hours long. This motivates the need for a new, open-sourced TTS model that can take handle tags from all categories, especially since the only open-source model, Parler-TTS, does not support any abstract tags at all. Second, all of the datasets that support abstract situational tags (emotions) only cover a maximum of 8 tags, motivating the need to substantially expand the space of abstract situational tags. Thirdly, while some of these datasets (e.g. TextrolSpeech) start with a limited set of tags (e.g. automatically extracted pitch, speaking rate and volume) and synthetically expand their style prompt vocabulary (e.g. by using LLMs to rewrite style tags with synonyms) to better mimic how humans would describe speech styles, this does not add a real signal to the dataset.

We resolve these issues by emphasizing our focus on a variety of abstract style tags that are difficult to extract using automatic extractors, requiring human annotations. We manually create a list of 47 abstract tags (26 intrinsic, 21 situational) covering pitch, texture, clarity, volume and rhythm for intrinsic and emotion and expressiveness for situational tags. Combined with 10 demographic tags (2 gender, 8 accents) and 6 automatic tags (3 pitch levels and 3 speaking rate levels), we cover a total of 63 tags; the full list is available in Appendix A. We set up 4 datasets aiming to target as many of these 63 tags as possible, and describe their creation in Section 3.

## 3 DATASETS

### 3.1 DATA COLLECTION

We present an overview of our dataset collection procedure for each tag category in Figure 1. We set up four datasets: (a) StyledVoxCeleb, a subset of the VoxCeleb (Nagrani et al., 2020) dataset we annotate with abstract intrinsic tags, (b) Expresso (Nguyen et al., 2023) and EARS (Richter et al., 2024), two existing expressive speech datasets whose speaking styles we remap to abstract situational tags, and (c) a 150-hr subset of LibriTTS-R (Koizumi et al., 2023) we call LTTSR-150 annotated with demographic and automatic tags. We annotate StyledVoxCeleb, Expresso and EARS with demographic and automatic tags as well. Preprocessing information for each dataset can be found in Appendix C. Across all datasets, every audio clip's style tag annotations are converted to a natural language style prompt using a Mistral (Jiang et al., 2023) LLM prompted with a comma-separated list of style tags and instructed to generate a style prompt (details in Appendix D). Every

| Dataset | # Spkr | # Utts. | Dur. |
|---|---|---|---|
| StyledVoxCeleb | 596 | 116k | 256.08h |
| Expresso | 4 | 16k | 30.21h |
| EARS | 107 | 15k | 60.58h |
| LTTSR-150 | 2410 | 95k | 178.52h |

Table 2: Dataset statistics.

example in our datasets thus consists of (a) an audio clip, (b) a text style prompt generated from the annotated style tags and (c) a text transcription.

ABSTRACT INTRINSIC TAGS    We create the StyledVoxCeleb dataset by annotating a subset of VoxCeleb (Nagrani et al., 2020) (a dataset consisting of natural, in-the-wild speech from YouTube celebrity interviews with high speaker diversity spanning accents, ages and ethnicities) with abstract intrinsic tags by hiring workers on Amazon Mechanical Turk. We apply this annotation to all utterances spoken by that speaker. This data collection effort is complementary to prior work (Kawamura et al., 2024) that collected such data for the LibriTTS-R (Koizumi et al., 2023) dataset. We show in Section 5 that our dataset outperforms LibriTTS-P when evaluated for speech-style consistency.

**Quality Control**    We provide a qualification task to Amazon Mechanical Turk workers to check their ability to understand style tags. The task consists of 6 manually selected pairs of speech clips where one exhibits a style and one doesn't. We ask annotators to select which one exhibits the style and keep only those 38 annotators that succeeded on at least 5 examples; details in Appendix E.

**Collecting Annotations**    Given a speaker, we create a representative audio file consisting of multiple utterances ($3 - 8$ clips whose total duration is $20 - 40$ seconds) concatenated together. We provide this audio file, the speaker's name and a list of our intrinsic speech style tags with definitions (see Appendix A) to annotators on Amazon Mechanical Turk and ask them to write at least 3 distinct style tags. Our annotation UI can be viewed at Appendix E. For every celebrity, we collect 5 annotations. We observe that the annotations are very subjective and different annotators select different tags for the same celebrity. Therefore, we keep only those tags that at least 2 annotators agree on in our train and dev set, and only those that at least 3 annotators agree on in our test set.

**Selecting Celebrities**    We expect famous or distinctive celebrities to be more familiar to annotators. We select such a subset using three loose heuristics: (a) we parse an IMDb list of 163 celebrities with distinctive voices [3] and find 39 in VoxCeleb, (b) we ask ChatGPT to name 300 celebrities with distinctive voices and find 112 in VoxCeleb, and (c) we find Wikipedia pages for VoxCeleb celebrity using the Python Wikipedia API [4] and select the top 200 celebrities by length of their Wikipedia pages, assuming page length is a proxy for fame. Combining all three sources and accounting for overlap, we obtain a list of 302 celebrities. After collecting annotations for these celebrities, we find that the style tag distribution is imbalanced, with 12 tags [5] having fewer than 5000 annotated clips. We use GPT-4 (OpenAI et al., 2024) to obtain a rough list of celebrities that are likely to have these tags by instructing it to output a list of style tags that are associated with a celebrity's voice (details in Appendix D) for every celebrity in VoxCeleb. Since we can only provide the celebrity's name rather than the actual speech clip, GPT-4 needs to rely on its parametric knowledge base in order to complete this task; while imperfect, it may still provide some signal towards which celebrities to target. We select a maximum of 40 celebrities per tag, ending up with a list of 187 additional celebrities to annotate (most tags have far fewer than 40 celebrities labelled by GPT-4). Finally, we annotate 107 additional celebrities, resulting in a total of 596 celebrities for StyledVoxCeleb.

We split every speaker in StyledVoxCeleb into train ($80\%$), dev ($10\%$), and test ($10\%$), ensuring that there is no transcript overlap across splits.

---

[3] https://www.imdb.com/list/ls001839542/

[4] https://github.com/martin-majlis/Wikipedia-API

[5] lisp, hushed, pitchy, staccato, monotonous, punctuated, vocal fry, guttural, singsong, soft, stammering, shrill

**ABSTRACT SITUATIONAL TAGS** We reuse Expresso (Nguyen et al., 2023) and EARS (Richter et al., 2024), two existing expressive speech datasets that consist of speakers acting out various emotions and speaking styles. Expresso contains 4 speakers while EARS contains 107. We filter out neutral and non-speaking utterances in both datasets, lightly preprocess them (details in Appendix C) and then label each utterance by simply mapping the reading styles in each dataset to our tag vocabulary (details in Table 6). For example, the *projected* style in Expresso gets mapped to the tag *loud*. We split the Expresso dataset into train (80%), dev (10%), and test (10%), ensuring that there is no transcript overlap across splits. Some utterances in EARS do not have emotion labels; we place them all into the train set. We split the utterances that have emotion labels into train (80%), dev (10%), and test (10%), ensuring overall that there is no transcript overlap across splits.

**DEMOGRAPHIC AND AUTOMATIC TAGS** We use either dataset metadata or GPT-4 for obtaining accent and gender, and automatic signal processing tools for extracting pitch and speaking rate. Following previous work (Lyth & King, 2024), we also extract the noise level of the audio (this is not a *style tag*; rather, it aids the model to differentiate between clean and noisy speech). We extract these tags for all 3 datasets: StyledVoxCeleb, Expresso and EARS. Additionally, we annotate a 150-hr subset of the train split of the LibriTTS-R (Koizumi et al., 2023) dataset (along with its dev and test sets) we call LTTSR-150 with gender, pitch, speaking rate and noise levels.

**Gender and Accent** For StyledVoxCeleb, we prompt GPT-4 with the name of the celebrity and ask it to output the celebrity's gender and accent. [6] The prompt and generation details are available in Appendix D. We use dataset metadata for Expresso, EARS and LibriTTS-R (for EARS, we use the 'native language' column of the dataset as a proxy for accent).

**Pitch, Speaking Rate and Noise Levels** We use the Dataspeech (Lacombe et al., 2024a) library to label our datasets with pitch, speaking rate, and noise levels. For pitch, we use PENN [7] using default hyperparameters and compute the mean pitch across all utterances of a given speaker. Then, we apply gender-dependent thresholds to label each speaker with low-pitched (male: $< 115.7$ Hz, female: $< 141.6$ Hz), high-pitched (male: $> 149.7$ Hz, female $> 184.5$ Hz) or medium-pitched (male: $115.7$ Hz $< x < 149.7$ Hz, female: $141.6$ Hz $< x < 184.5$ Hz); these thresholds are gender-dependent since humans perceive male speakers to have lower pitch than female speakers on average. For speaking rate, we use g2p [8] to convert the text transcription to phoneme transcriptions and then use the number of phonemes per second (PPS) as the speaking rate. We apply thresholds to label each utterance with slow ($< 11.5$ PPS), fast ($> 19.1$ PPS) and measured ($11.5$ PPS $< x < 19.1$ PPS). Finally, for noise levels, we use the signal-to-noise ratio (SNR) extracted using Brouhaha [9] and use Parler-TTS (Lacombe et al., 2024b)'s noise bins to assign each utterance one of the following noise tags: *very noisy, quite noisy, slightly noisy, moderate ambient sound, slightly clear, quite clear, very clear*.

## 3.2 DATASET STATISTICS

We report dataset statistics for each dataset we setup (StyledVoxCeleb, Expresso, EARS and the LibriTTS-R subset LTTSR-150) combining train, dev and test splits in Table 2. We report the distribution of each accent tag in Figure 2 and abstract tags in Figure 3. The distribution of gender tags is 56.6% male, 43.4% female, of pitch tags is 21.4% low-pitched, 41.5% medium-pitched and 37.05% high-pitched, and of speaking rate tags is 12.6% slow, 75.7% measured and 11.6% fast.

## 4 EXPERIMENTAL SETUP

We use the Parler-TTS (Lyth & King, 2024; Lacombe et al., 2024b) model as our backbone for all experiments. Parler-TTS is a style-prompted TTS model trained on 45K hours of data, consisting of the English split of Multilingual Librispeech (Pratap et al., 2020) and LibriTTS-R (Koizumi et al., 2023) annotated with automatic style tags.

---

[6]We manually verified a subset of the generated metadata and found it to be of high quality.

[7]https://github.com/interactiveaudiolab/penn

[8]https://github.com/roedoejet/g2p

[9]https://github.com/marianne-m/brouhaha-vad

**Model Architecture**  The crux of Parler-TTS is an autoregressive decoder speech language model that generates DAC (Kumar et al., 2023) audio tokens. To condition on text transcripts, the text transcript is tokenized using the Flan-T5 (Chung et al., 2022) tokenizer, passed through a linear embedding layer, and prepended to the input sequence of the decoder. To condition on the text style prompt, the text encoder, a frozen Flan-T5 model, maps the text style prompt to a sequence of hidden-state representations that are attended to via cross-attention layers in the decoder.

**Training**  We initialize our model with the `parler-tts/parler-tts-mini-v1` open-source checkpoint and use the official Parler-TTS [10] library to finetune on the training splits of the 4 datasets we set up; StyledVoxCeleb, Expresso, EARS and LTTSR-150. We train on 4 NVIDIA A40 GPUs with a batch size of 8 and 2 gradient accumulation steps. We train for 9 epochs with a non-warmup cosine learning rate scheduler, a peak learning rate of 0.00008 and a weight decay of 0.01.

**Inference**  We perform inference runs using the default Parler-TTS generation hyperparameters (temperature 1.0, repetition penalty 1.0, 2580 total tokens). Since autoregressive TTS is prone to decoding instabilities, we attempt to mitigate this by retrying inference a maximum of 3 times, stopping when the WER between the ASR transcript of the generated sample and the input text (using the same setup as our WER evaluation metric) falls below 20 or choosing the sample with the lowest WER out of the 3 generated samples.

## 4.1 EVALUATION DATASET

We start by combining the test splits of StyledVoxCeleb, Expresso, EARS and LibriTTS-R. For each tag in our tag vocabulary, we find a maximum of 5 clips that have been annotated with that tag and select them for inclusion in our evaluation dataset. For each clip, we refer to this tag as its *tag of interest*. We randomly select pitch and speaking rate with a 50% probability for inclusion along with the tag of interest, gender, noise level and then generate a style prompt from these tags for use in evaluation. Our final evaluation dataset consists of 298 clips.

## 4.2 EVALUATION METRICS

We use metrics that evaluate the speech clip for three desiderata: speech quality, content correctness and speech-style consistency. For human evaluation metrics, we use annotators recruited on Amazon Mechanical Turk; Appendix E contains details about our annotation user interfaces and annotation costs. For every human evaluation metric, we collect 3 human annotation scores per test dataset item. We report the mean and 95% confidence intervals of the MOS scores (Ribeiro et al., 2011).

**Speech Quality**  Following previous work (Vyas et al., 2023; Kawamura et al., 2024), we compute a Naturalness MOS metric where each human annotator is provided speech clips and asked to rate its naturalness and realisticity (human-likeness) on 5-point Likert scales.

**Content Correctness**  We report a WER metric that computes the Word Error Rate (WER) between (a) the ASR transcript of the speech clip and (b) the input transcript, after applying a text normalizer to both texts. We use the distil-whisper/distil-large-v2 (Gandhi et al., 2023) model for ASR, and Whisper[11] for text normalization.

**Speech-Style Consistency**  Following Kawamura et al. (2024) and Ji et al. (2024), we report a Consistency MOS metric where each human annotator is provided a speech clip and the input style prompt and asked to rate the consistency between the two on a 5-point Likert scale.

In addition, we report fine-grained tag-level evaluation. Instead of evaluating adherence to the whole style prompt (e.g., *A woman's speech is delivered slowly with a high-pitched tone, expressing disgusted emotions, in a clear and quiet environment*), we ask annotator to select style tags they hear. For example, for the same example, annotator might select *female, disgusted* as pronounced style.

---

[10] https://github.com/huggingface/parler-tts

[11] https://github.com/huggingface/transformers/blob/main/src/transformers/models/whisper/english_normalizer.py

For each tag, we compute its recall (fraction of instances in which the relevant tag was selected by the annotator), and report the average tag recall as well as per-category average tag recall.

For two automatic tag types (pitch and speaking rate), we further report an Accuracy score. We run the generated speech clip through the same pitch and speaking rate extractors used to build our datasets to obtain predicted style tags. We use the style prompt's gender to decide which pitch bins to use and label each generated utterance individually, rather than speaker-level mean aggregation used for building the datasets. We compute the speaking rate from the phoneme sequence obtained from the ASR transcript of the generated speech. We then compare the predicted tags pitch with the desired tags in the input style prompt, giving a score of 1 if the labels match and 0 otherwise.

### 4.3 BASELINES

Due to the absence of open-source style-prompted TTS models other than Parler-TTS, all our baselines finetune Parler-TTS on different datasets with the same training and inference setup as ours.

**Init.** This is the Parler-TTS model that we initialize all models with.

**+LTTSR** We finetune Parler-TTS on the LibriTTS-R (Koizumi et al., 2023) dataset. We extract gender tags using dataset metadata and automatic tags using our signal processing pipeline for extracting pitch, speaking rate and noise levels. While Parler-TTS is already trained on LibriTTS-R, it uses different binning thresholds for pitch and speaking rate; this baseline ablates that mismatch.

**+LTTSP,Exp,EARS** We finetune Parler-TTS on a combination of existing datasets that cover all tag categories: Expresso and EARS for abstract situational tags and LibriTTS-P (Kawamura et al., 2024) which annotates the LibriTTS-R dataset with abstract intrinsic tags. LibriTTS-P provides 3 annotations (each consisting of a list of style tags) per speaker and each style tag optionally has one of two qualifiers (*slightly* and *very*) that indicates the strength of the style tag. We preprocess the annotations by removing tags with the *slightly* qualifier and remapping some style tags to those in our vocabulary (see Appendix C). For each clip in the dataset, we select one of the three annotations corresponding to its speaker at random and combine with automatic tags extracted using our pipeline. This baseline ablates the use of LibriTTS-P versus our StyledVoxCeleb dataset.

## 5 RESULTS

| Model | Cons. MOS ↑ | Tag Recall ↑ | | | | | Accuracy % ↑ | |
|---|---|---|---|---|---|---|---|---|
| | | **All** | **Intr** | **Sit** | **Dem** | **Auto** | **Pitch** | **Rate** |
| GT | $3.76 \pm 0.57$ | 0.62 | 0.56 | 0.62 | 0.74 | 0.70 | 56.52 | 93.24 |
| Init. | $3.14 \pm 0.44$ | 0.25 | 0.23 | 0.16 | 0.28 | **0.59** | 62.73 | **77.01** |
| +LTTSR | $3.15 \pm 0.47$ | 0.26 | 0.20 | 0.19 | 0.33 | 0.58 | **73.91** | 66.21 |
| +LTTSP,Exp,EARS | $3.19 \pm 0.31$ | 0.30 | 0.24 | **0.29** | 0.26 | **0.60** | 72.67 | 75.00 |
| Ours | $\mathbf{3.29 \pm 0.40}$ | **0.36** | **0.29** | **0.29** | **0.52** | **0.61** | 72.05 | 75.68 |

Table 3: Speech-Style Consistency results comparing various baseline models and ours. We report the mean and 95% confidence intervals for Consistency MOS. Tag recalls are averaged across all tags (All) and broken down by each tag category (Intr. is abstract intrinsic, Sit. is abstract situational, Demo. is demographic and Auto. is automatic). We find that our model outperforms baselines at consistency MOS and overall Tag Recall.

**Speech-Style Consistency** Table 3 reports model performance along various metrics that aim to evaluate how well the generated speech adheres to the provided text style prompt. The consistency MOS ranges from $1 - 5$, the tag recall from $0 - 1$ and the accuracies from $0 - 100\%$. Our model achieves the highest consistency MOS score, verified by running a paired bootstrap significance test comparing the two highest MOS scores (ours and the +LTTSP,Exp,EARS baseline) that finds the difference is statistically significant with a p-value of 0.004. Furthermore, the Tag Recall

| Model | NMOS ↑ | WER ↓ |
|-------|--------|-------|
| GT | $3.94 \pm 0.42$ | 8.10 |
| Init. | $3.05 \pm 0.25$ | 4.91 |
| +LTTSR | $\mathbf{3.12} \pm 0.20$ | **4.75** |
| +LTTSP,Exp,EARS | $2.99 \pm 0.18$ | 6.33 |
| Ours | $2.80 \pm 0.16$ | 9.12 |

Table 4: Speech Quality and Content Correctness results. We report the mean and 95% confidence intervals for Naturalness MOS.

scores provide a more finegrained understanding of model performance. We outperform all baselines, including the LTTSP,Exp,EARS baseline on intrinsic tags, showing the benefits of training on StyledVoxCeleb versus LibriTTS-P. Since both our model and the LTTSP,Exp,EARS baseline is trained on Expresso and EARS, we match performance on situational tags, but outperform other baselines, demonstrating the benefits of training with Expresso and EARS. Additionally, our model outperforms on demographic tags as well due to the presence of a rich diversity of accents in StyledVoxCeleb. When automatic tags (pitch and speaking rate) are evaluated via automatic accuracy scores, we find that the baselines trained without any abstract tags (Init. and +LTTSR) slightly outperform our model by about 2%. However, all models perform similarly at automatic tags when evaluated using tag recall, showing that humans do not exhibit strong preferences between models when evaluating pitch or speaking rate. We note that the ground truth pitch accuracy is unusually low because of a mismatch between how pitch is computed during evaluation versus dataset construction: the pitch is computed on an utterance-level basis during evaluation, while it was obtained on a speaker-level basis when constructing the style prompt in the dataset.

**Speech Quality and Content Correctness**    Table 4 compares the naturalness and content correctness of the generated speech across models. We find that the models trained without any abstract tags (+LTTSR and Init.) widely outperform our model and the +LTTSP,Exp,EARS baseline on both naturalness and WER. Training on LibriTTS-P, Expresso and EARS, despite being clean, high-quality audio data worsens both WER and naturalness. We hypothesize this is due to the introduction of speaking styles that are harder to transcribe and the relatively small scale of the Expresso and EARS data. Furthermore, training on StyledVoxCeleb (Ours) worsens it further, which we hypothesize is due to the presence of noisier in-the-wild speech (VoxCeleb) in our training data, which introduces speech artifacts in the generated speech. This is a limitation of the audio quality and size of our dataset, which we expect will be mitigated as we scale our dataset to more speakers (for example, Voicecraft (Peng et al., 2024), a voice cloning TTS model trains on large-scale, in-the-wild noisy data and achieves low Word Error Rates). The WER of Init. and LTTSR is substantially lower than the ground truth; this is likely because both models are trained on read audiobook data which is easier for humans and ASR systems to understand.

## 6    RELATED WORK

**Style-Prompted Text-to-Speech Models**    Table 1 already compares several existing style-prompted text-to-speech papers with respect to our tag categorizations. PromptTTS (Guo et al., 2022), one of the first papers to introduce style-prompted TTS, consists of 4 emotions and automatic tags and is trained on a synthetic emotion dataset; PromptTTS2 (Leng et al., 2023), a successor focuses on an improved model architecture. Other emotion-focused work includes InstructTTS (Yang et al., 2023), PromptStyle (Liu et al., 2023) and MEAD-TTS (Guan et al., 2024) which focus on collecting or annotating emotional data using human voice actors or annotators. TextrolSpeech (Ji et al., 2024) also focuses on emotion by collating several existing emotion classification datasets for use for style-prompted TTS. Recently, Parler-TTS (Lacombe et al., 2024b; Lyth & King, 2024), AudioBox (Vyas et al., 2023) and SpeechCraft (Jin et al., 2024) proposed scaling up style-prompted TTS to a much larger pool of data; while Parler-TTS and SpeechCraft did so solely using automatic tagging pipelines, AudioBox used a combination of automatically tagged data and internally annotated stylistic datasets to train the model. LibriTTS-P (Kawamura et al., 2024) explores abstract

intrinsic tag annotations by annotating speakers in LibriTTS-R (Koizumi et al., 2023). Other relevant, contemporaneous style-prompted TTS work includes Chen et al. (2024); Zhu et al. (2024); Yamamoto et al. (2024).

**Style Control for other Speech Tasks**  Recent work has explored natural language style prompts for tasks other than TTS. DreamVoice (Hai et al., 2024), like LibriTTS-P, annotates LibriTTS-R with abstract intrinsic tags, but for the task of voice conversion. The contemporaneous VCTK-RVA (Sheng et al., 2024) annotates the VCTK dataset with intrinsic tags for training a speech editing system that conditions on a style prompt instruction.

## 7  CONCLUSION

We propose a crisper definition of speech style tags, categorizing into abstract intrinsic, abstract situational, demographic and automatic tags. We use this to substantially expand the space of style prompts by supporting 63 total tags. Emphasizing the importance of abstract tags, we collect intrinsic tag human annotations for a subset of speakers in the VoxCeleb (Nagrani et al., 2020) dataset to create StyledVoxCeleb, and reuse Expresso (Nguyen et al., 2023) and EARS (Richter et al., 2024) for situational tags. We train style-prompted TTS models based on Parler-TTS (Lacombe et al., 2024b; Lyth & King, 2024) that show improved performance on speech-style consistency metrics compared to competitive baselines, while they underperform baselines on speech quality and content correctness metrics.

## 8  LIMITATIONS

**Expensive human annotation**  Our dataset collection strategy relies on expensive, slow human annotation for abstract intrinsic and situational tags. While it is significantly cheaper to annotate intrinsic tags on a speaker level rather than situational tags on an utterance level, it is unclear how to substantially and cheaply scale either type of annotation. Future work could potentially look into using synthetic data augmentation (Défossez et al., 2024) for automatically expanding existing annotated datasets.

**Noisy data**  While VoxCeleb is beneficial as it has a high diversity of speakers and is sourced from a more realistic, in-the-wild speech domain, it negatively affects model performance when evaluated for speech quality and content correctness due to inherent background noise in a majority of its utterances. While this affects our models that are trained on a subset of VoxCeleb, we expect that scaling to more speakers will mitigate this issue to some extent.

**Language coverage**  We limit our current experiments to English data; there is a lot of potential to expand style-prompted TTS to more languages, both in terms of the language of the utterance and the language of the style prompt. Some work (Jin et al., 2024; Yamamoto et al., 2024) explores other languages like Chinese and Japanese in addition to English for style-prompted TTS.

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

## A    LIST OF SPEECH STYLE TAGS

This is the list of tags we consider:

- **Intrinsic:**
  - **Abstract:**
    * **Pitch**: Shrill, Nasal, Deep.
    * **Texture:** Silky, Husky, Raspy, Guttural, Vocal-fry.
    * **Clarity:** Crisp, Slurred, Lisp, Stammering.
    * **Volume:** Booming, Authoritative, Loud, Hushed, Soft.
    * **Rhythm:** Pitchy, Flowing, Monotonous, Staccato, Punctuated, Hesitant, Singsong, Enunciated.
  - **Demographic:**
    * **Gender:** Male, Female.
    * **Accent:** American, British, Scottish, Canadian, Australian, Irish, Indian, Jamaican.
  - **Automatic:**
    * **Pitch Levels:** High-pitched, Medium-pitched, Low-pitched.
- **Situational:**
  - **Abstract:**
    * **Emotion:** Enthusiastic, Happy, Angry, Saddened, Awed, Calm, Anxious, Disgusted, Scared, Confused, Bored, Sleepy, Pained, Guilt, Sarcastic, Sympathetic, Admiring, Desirous.
    * **Expressiveness:** Animated, Laughing, Passive, Whispered.
  - **Automatic:**
    * **Speaking Rate Levels:** Fast, Measured, Slow.

We note that our datasets contain more accent tags than those shown here, but these 8 accents are most represented in our datasets, and hence we evaluate on only these accents (accent distribution of our datasets can be viewed at Figure 2 Some tag categories like volume, speaking rate and rhythm can span both intrinsic and situational; however, we collect data for volume with intrinsic human annotations, and automatically obtain speaking rate tags on an utterance-by-utterance basis i.e. in a situational manner. Therefore, we place them in their respective intrinsic or situational categories. We collect all rhythm tags with intrinsic annotations, and place them in the intrinsic category; however, 2 rhythm tags (*singsong, enunciated* are also present in our situational datasets which we also use. The manually written definitions for each style tag can be found in Table 5.

## B    DATASET STATISTICS

We report dataset statistics for accent distribution in Figure 2 and abstract tags distribution in Figure 3.

## C    DATASET PREPROCESSING

For all datasets, we convert audio from their original format to the `.wav` format, apply loudness normalization using SoX and PyDub [12] such that the peak volume of each audio is $-0.1$ dB, and discard all audios shorter than 2 s or longer than 30 s. If an utterance does not come with ground truth transcripts, we synthesize transcripts using the Whisper (Radford et al., 2022) `large-v3` ASR model. We describe dataset-specific preprocessing below:

---

[12]`https://sourceforge.net/projects/sox/`, `https://github.com/jiaaro/pydub`

| Attribute | Description |
| --- | --- |
| High-pitched | A voice with a distinctly high frequency. |
| Shrill | A high-pitched, piercing, and sharp voice. |
| Nasal | A whiny voice that sounds like someone is speaking through their nose. |
| Medium-pitched | A voice with a medium frequency that is neither very high or low-pitched. |
| Low-pitched | A voice with a distinctly low frequency. |
| Deep | A low-pitched, resonant, rich voice. |
| Silky | A smooth, pleasant and soothingly soft voice. |
| Husky | A slightly rough, low voice that conveys a gritty texture. |
| Raspy | A rough, grating, somewhat harsh voice. |
| Guttural | A deep, throaty, gravelly voice. |
| Vocal-fry | A creaky, breathy voice that occurs when vocal cords flutter and produce a sizzling, popping sound at ends of sentences. |
| American | A voice with an American accent. |
| British | A voice with a British accent. |
| Scottish | A voice with a Scottish accent. |
| Canadian | A voice with a Canadian accent. |
| Australian | A voice with a Australian accent. |
| Irish | A voice with an Irish accent. |
| Indian | A voice with an Indian accent. |
| Jamaican | A voice with a Jamaican accent. |
| Male | A male voice, often having a lower pitch. |
| Female | A female voice, often having a higher pitch. |
| Booming | A loud, resonant, commanding, powerful voice. |
| Authoritative | A confident, clear voice with a tone that conveys expertise and assurance. |
| Loud | A voice with a high volume. |
| Hushed | A soft, quiet, low-volume voice typically used to convey intimacy or secrecy. |
| Soft | A gentle, low-volume, calm and soothing voice typically used to convey subtlety. |
| Whispered | A breathy, low-volume voice typically used to speak discreetly. |
| Crisp | A clear, distinct, articulate voice. |
| Slurred | An unclear, difficult-to-understand voice that blends together sounds and words. |
| Lisp | A speech pattern that involves difficulty in speaking certain consonants e.g. 's' and 'z' are spoken with a 'th' sound. |
| Stammering | A voice with pauses, repetitions and prolongations of words that disrupt the speech flow. |
| Singsong | A melodious voice that rises and falls in a musical manner. |
| Pitchy | A jarring, somewhat unstable voice that often strays from the correct pitch. |
| Flowing | A clear, coherent, seamless and easy-to-understand voice. |
| Monotonous | A dull, flat voice whose pitch, tone and speed remains constant throughout. |
| Staccato | A disjointed, unclear voice with breaks in-between syllables or words. |
| Punctuated | An engaging voice with clear, deliberate pauses that emphasize key words. |
| Enunciated | A voice that clearly and precisely articulates words, with each syllable distinctly pronounced. |
| Fast speed | A rapidly speaking, quick voice with few pauses. |
| Measured speed | A controlled, deliberate voice that has an even tone and a moderate speed. |
| Slow speed | A voice with a slower speaking rate. |
| Hesitant | An uncertain, tentative voice, often marking a lack of confidence, reluctance or confusion. |
| Enthusiastic | A lively, energetic, positive voice that conveys excitement and interest in the topic being discussed. |
| Happy | A warm, positive and joyful voice. |
| Angry | A raised voice that conveys anger, frustration or displeasure, characterized by raised volume and emphatic speech patterns. |
| Saddened | A voice with a low, subdued, and unenergetic tone that conveys distress, disappointment or sadness. |
| Awed | A voice that conveys the speaker's admiration, wonder or reverance of something the speaker appreciates. |
| Calm | A calm, gentle and serene voice that conveys the speaker's relaxed and peaceful emotion. |
| Anxious | A voice that conveys nervousness and anxiety, often marked by rapid or jittery speech patterns. |
| Disgusted | An intonated voice that conveys repulsion and disgust by appropriately altering its pitch and rhythm. |
| Scared | A shaky, rapid voice that reflects the speaker's anxiety or fear. |
| Confused | A voice characterized by indecision and a lack of clarity, often marked by hesitance. |
| Bored | A voice, often monotonous, that indicates lack of enthusiasm and disinterest. |
| Sleepy | A soft, slow, low-energy voice that indicates tiredness. |
| Pained | A voice characterized by a strained, trembling tone that indicates sorrow or anguish. |
| Guilt | A voice that carries a wavering, hesitant tone that hints at discomfort or regret. |
| Sarcastic | A speaking style that is characterized by a distinct tone of irony that suggests that the speaker's intention is to mock or convey contempt. |
| Sympathetic | A gentle, compassionate voice that reassures and seeks to empathize with the listener. |
| Admiring | An appreciative, positive and complimentary manner of speaking. |
| Desirous | An emotional voice that conveys deep longing or desire. |
| Animated | A energetic, heightened voice characterized by varied intonations or emotional intensity. |
| Laughing | A voice with intermittent sounds of laughter conveying amusement and joy. |
| Passive | A tentative, subdued and uninterested voice. |

Table 5: Manually written style tag definitions.

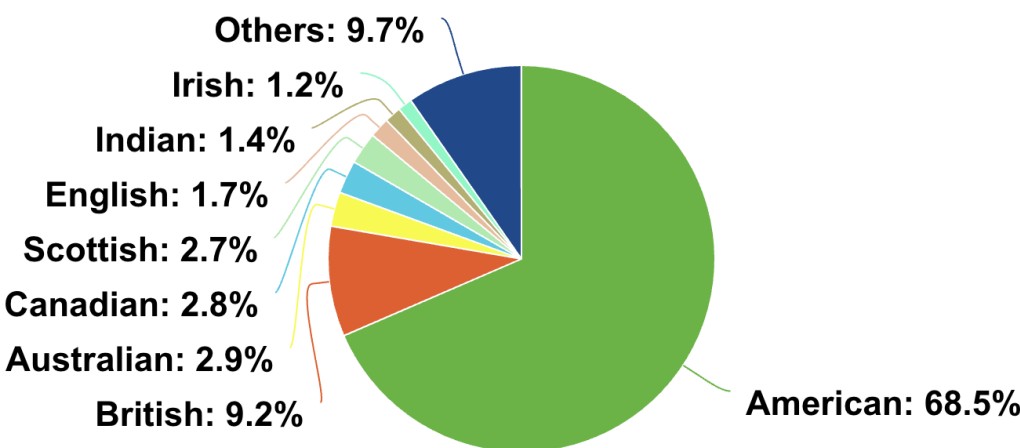

Figure 2: Accent distribution across datasets.

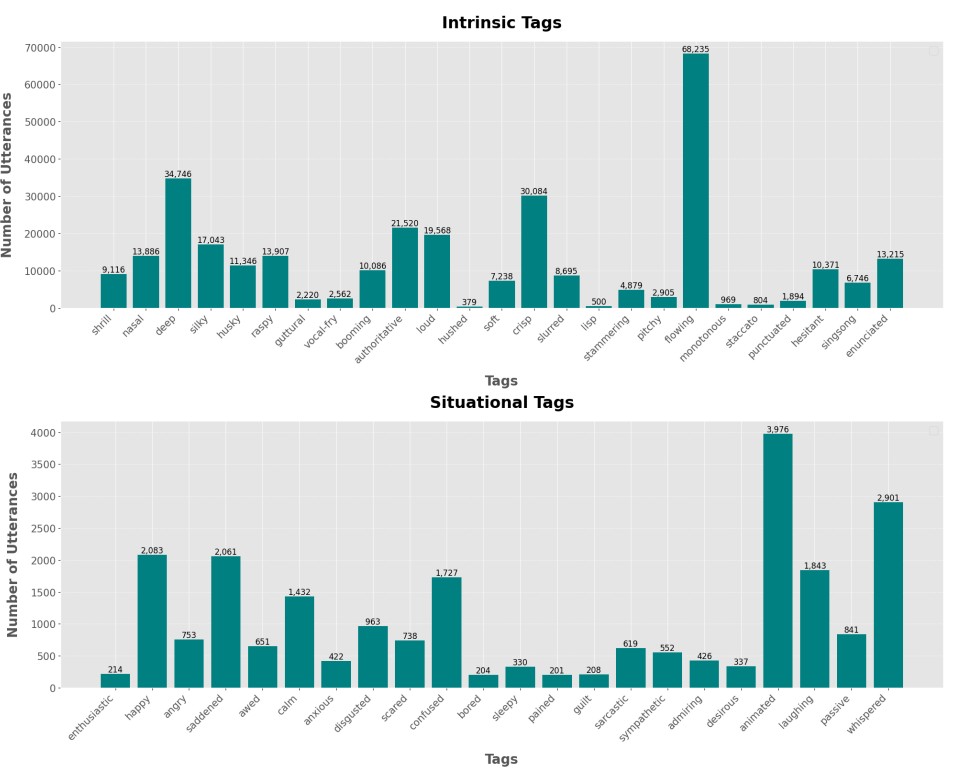

Figure 3: Histogram of abstract tag distribution across datasets.

## C.1 STYLEDVOXCELEB

We combine the VoxCeleb1 and VoxCeleb2 datasets. We apply a noise removal model, Voice-fixer (Liu et al., 2021) to all audios, since we observed that a significant proportion of VoxCeleb data is noisy (the median SNR for VoxCeleb data is 31.76 dB computed by Brouhaha (Lavechin et al., 2023); compare to 59.49, 50.42 and 61.70 for Expresso, EARS and LibriTTS-R respectively). We then run a language identification model Lingua [13] over the transcripts and only keep those examples whose transcripts are identified as English text and discard celebrities with fewer than 10 English audio clips.

---

[13] https://github.com/pemistahl/lingua-py

## C.2 EXPRESSO

The Expresso dataset consists of 4 voice actors speaking various utterances in different speaking styles. We discard the *default, narration* and *non-verbal* speaking styles, since they do not exhibit the situational tags we are interested in. Since some of the data is in the form of long freeform dual-channel conversations between two voice actors, we use the Voice Activity Detection data provided by the dataset to splice the long conversation into two channels and VAD-segmented chunks, so that we can use each chunk as an utterance. We then remap each speaking style to our tag vocabulary as described in Section C.4.

## C.3 EARS

The EARS dataset consists of 107 speaking enacting various speaking styles. We discard the long *freeform* examples as they are not labelled with speaking styles. We also discard *interjection, nonverbal* and *vegetative* speaking styles since they do not contain natural speech. We remap the speaking styles in the rest of the data to our tag vocabulary as described in Section C.4.

## C.4 TERM REMAPPING

We remap terms in the Expresso, EARS, and LibriTTS-P datasets to terms in our vocabulary using the mapping in Table 6.

| Original Term | Mapped Term(s) | Original Term | Mapped Term(s) |
|---|---|---|---|
| feminine | female | awe | awed |
| halting | stammering | bored | bored, passive |
| tensed | anxious | desire | desirous, animated |
| relaxed | calm | projected | loud |
| powerful | authoritative | fearful | scared |
| muffled | slurred | amusement | happy |
| masculine | male | distress | anxious, scared |
| fluent | flowing | disappointment | saddened, passive |
| weak | hushed | realization | awed |
| sharp | crisp | amazement | awed |
| reassuring | sympathetic | disgust | disgusted |
| lively | enthusiastic | fear | scared |
| cool | calm | anger | angry |
| happy | happy, animated | adoration | admiring |
| laughing | laughing, animated | confusion | confused |
| sad | saddened | interest | enthusiastic |
| whisper | whispered | serenity | calm |
| singing | singsong | contentment | calm, passive |
| angry | angry, animated | sadness | saddened |
| desire | desirous | extasy | happy |
| interest | enthusiastic | pain | pained |
| serenity | calm | cuteness | happy |
| contentment | calm, passive | relief | calm, passive |
| sadness | saddened | pride | admiring |
| loud | loud | embarrassment | anxious |
| whisper | whispered | | |

Table 6: Terms in existing datasets remapped to terms in our vocabulary.

## D    LLM PROMPTING

### D.1    IMPERFECTLY LABELLING CELEBRITIES WITH STYLE TAGS

We use the `gpt-4-0125-preview` version of GPT-4 via the OpenAI API with the default hyperparameters (temperature 1.0, top-p 1.0, maximum 2048 tokens). We prompt it with the name of the celebrity and ask it to output a list of style tags associated with the celebrity's voice with the following prompt template:

```
Given the name of a famous celebrity or actor, you must retrieve
↪   your knowledge about that celebrity's voice and map the voice
↪   to a subset of speech style attribute labels provided to you.
↪   Here is the list of speech style attribute types you should
↪   pay attention to, along with attribute labels for each type:
<attributes>
- **Pitch:** Shrill, Nasal, Deep.
- **Texture:** Silky, Husky, Raspy, Guttural, Vocal-fry.
- **Volume:** Booming, Authoritative, Loud, Hushed, Soft.
- **Clarity:** Crisp, Slurred, Lisp, Stammering.
- **Rhythm:** Singsong, Pitchy, Flowing, Monotonous, Staccato,
↪   Punctuated, Enunciated, Hesitant.
</attributes>

Your task is to associate the celebrity with a subset of these
↪   attributes, taking into account how the celebrity always
↪   sounds like. Only use the attributes that are extremely
↪   salient to the celebrity's voice i.e. their unique speech
↪   styles. Don't create any new attributes because you will fail
↪   the task if you do so.

The celebrity is {name}. First generate a paragraph of around 5
↪   sentences, within <description> tags, using your knowledge,
↪   that describes the salient attributes of {name}'s voice. Then,
↪   within <attribute> tags, generate a list of comma-separated
↪   speech style attributes, from the above attributes list, that
↪   saliently apply to {name}. Use the following format:
<description>
(Description goes here)
</description>
<attribute>
(Comma-separated list of attributes)
</attribute>
```

### D.2    EXTRACTING GENDER AND ACCENT

We use the `gpt-4-0125-preview` version of GPT-4 via the OpenAI API with the default hyperparameters (temperature 1.0, top-p 1.0, maximum 2048 tokens). We prompt it with the name of the celebrity and ask it to output the celebrity's gender and accent with the following prompt template:

```
Tell me the accent and the gender of {name} formatted as
Accent: <accent>
Gender: <gender>
```

### D.3    GENERATING STYLE PROMPTS

We use the Mistral-7B-Instruct-v0.2 LLM (Jiang et al., 2023) to generate prompts via the Dataspeech library with a per-device batch size of 32 and sample with a temperature of 0.6, a top-p of 1.0 with a maximum 256 new tokens. We prompt the model with a comma-separated list of style tags and instruct it to generate a style prompt with the following prompt:

```
An audio sample of a person's speech can be described in several
↪   ways using descriptive keywords. These keywords may include
↪   demographic data about the person (e.g. gender, name, accent)
↪   and voice characteristics (e.g. related to pitch, gender,
↪   texture and rhythm, volume, clarity, speaking rate, emotion,
↪   expressiveness).

You will be provided several keywords that describe the speech
↪   sample. Your task is to create a simple text description using
↪   the provided keywords that accurately describes the speech
↪   sample. Ensure that the description remains grammatically
↪   correct, easy to understand, and concise. You can rearrange
↪   the keyword order as necessary, and substitute synonymous
↪   terms where appropriate. After you are provided the keywords,
↪   generate only the description and do not output anything else.

An example is provided below.
female, confused, hesitant, slightly noisy environment

Description: A woman's speech sounds confused and hesitant,
↪   recorded in a slightly noisy environment.

Now, generate a description for the following example:
{all_tags_str}

Description:
```

# E  HUMAN ANNOTATION: DETAILS

## E.1  ANNOTATION DETAILS

We recruit Amazon Mechanical Turk workers certified as Verified workers with a minimum approval rate of 98% and at least 500 successful HITs. We perform a qualification task using 6 pairs of manually selected clips from VoxCeleb or Expresso where one clip exhibits a style (one of *deep, whispered, scared, slurred, high-pitched, enunciated*) and the other doesn't, and select those 38 annotators that succeed in finding the right clip for at least 5 of the 6 pairs. We use this pool of annotators for our data collection. For evaluation metrics, we use all Verified workers with a minimum approval rate of 98% and at least 500 successful HITs rather than just our pool of 38 workers for faster evaluation turnaround. We pay annotators $9/hr.

## E.2  ANNOTATION USER INTERFACES

We display the annotation UIs for qualification task in Figure 4, crowdsourcing abstract intrinsic style tag annotations in Figure 5, speech quality evaluation in Figure 6, and speech-style consistency evaluation in Figure 7.

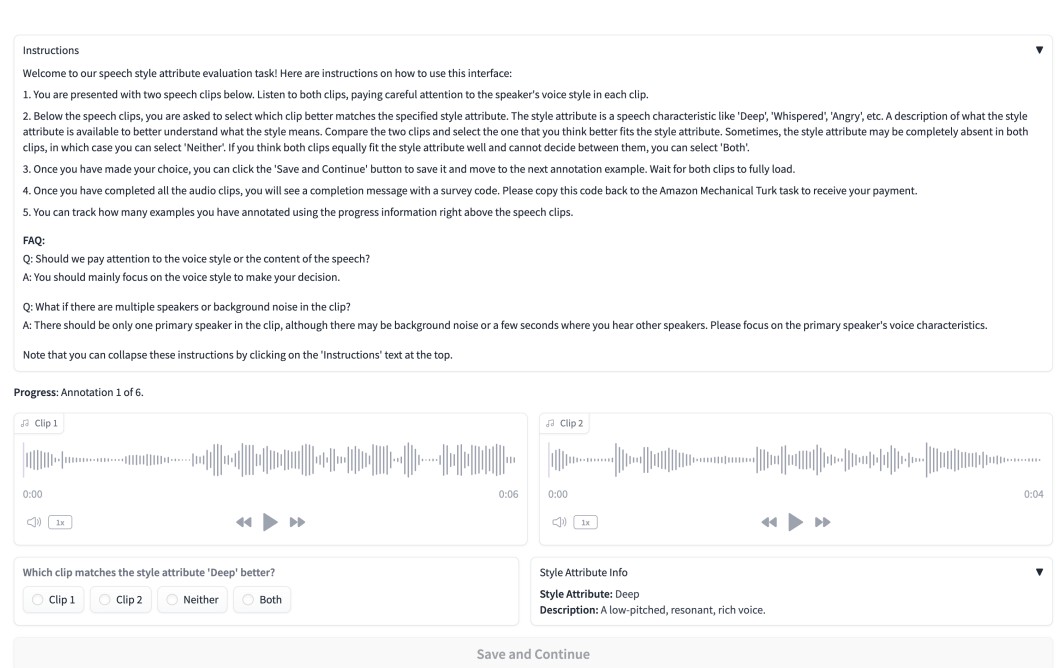

Figure 4: Annotation UI for selecting qualified annotators.

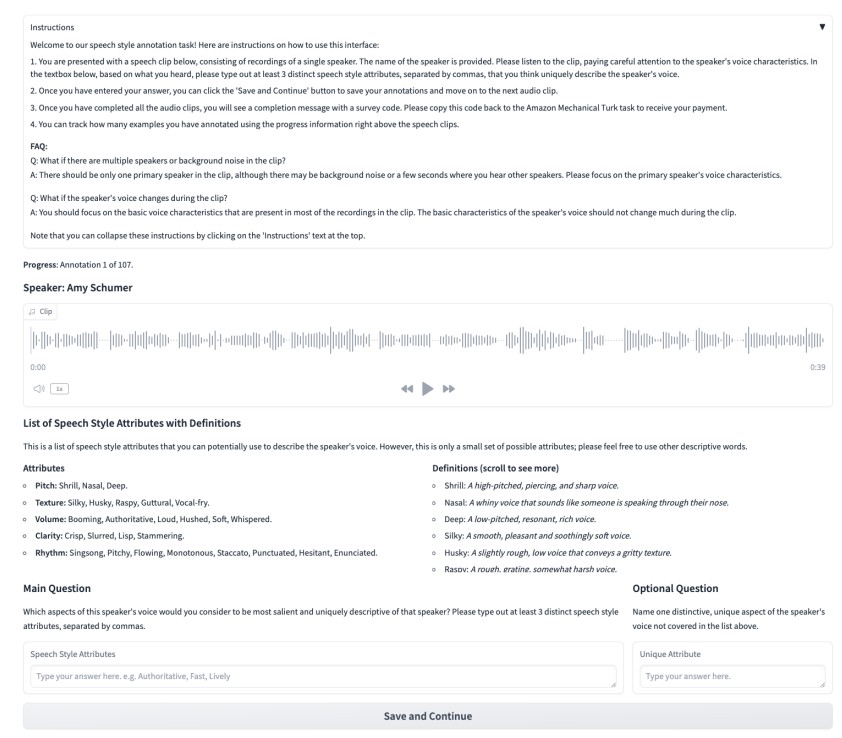

Figure 5: Annotation UI for crowdsourcing abstract intrinsic style tag annotations.

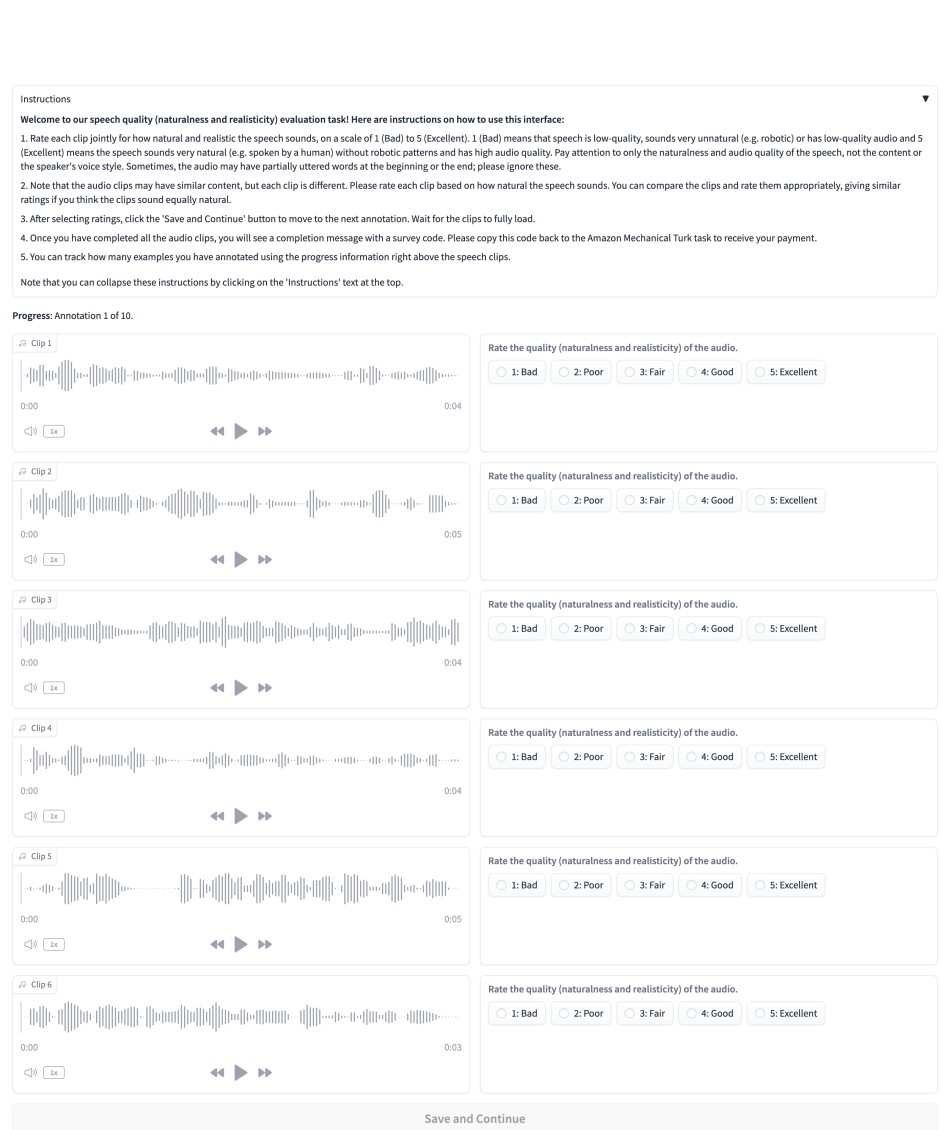

Figure 6: Annotation UI for collecting Naturalness Mean Opinion Score ratings.

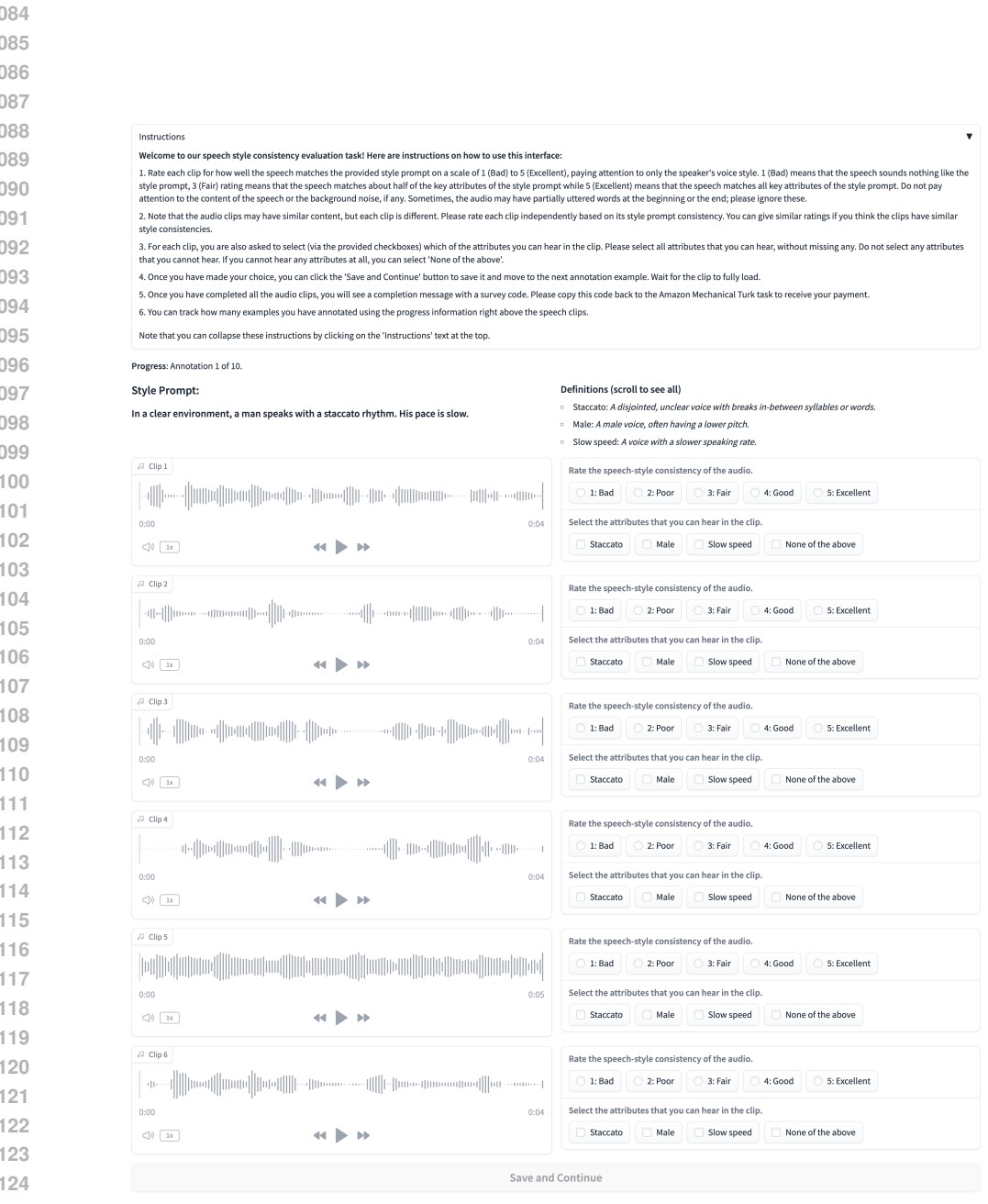

Figure 7: Annotation UI for collecting Consistency Mean Opinion Score and Tag Recall ratings.

