# OpenReview forum: "Modeling Abstract Style Prompts for Text-to-Speech Models"
_ICLR.cc/2025/Conference — ICLR 2025 Conference Withdrawn Submission_

### Official Review · Reviewer_gVmW · 2024-10-25

**Soundness:** 3
**Presentation:** 4
**Contribution:** 3
**Rating:** 6
**Confidence:** 4

**Summary:**

Although style-prompted TTS systems typically employ style tags to automatically constrcuct textual style prompts data, there has been a lack of systematic discussion on these tags. To address this gap, this work presents a systematic categorization of style tags across two dimensions. Compared with previous datasets, this work incorporates a broader range of style tags and annotates a 200-hour subset of the VoxCeleb dataset, focusing on tags that were previously underrepresented. The experiments on an open-source style-prompted TTS system validates the effectiveness of the proposed dataset.

**Strengths:**

- The annotated dataset and model weights will be open-sourced.

- The categorization of style tags is beneficial for expanding the scope of text prompts constructed based on tags, thereby facilitating better generalization to style prompts encountered in real-world scenarios.

**Weaknesses:**

- The baseline model exhibits noticeable instability, with the paper suggesting that the WER could surpass 20%. Such instability may substantially affect the assessment of stylistic elements.

- It may be inaccurate to claim that "autoregressive TTS systems are inherently prone to decoding instabilities" , as evidenced by robust autoregressive TTS systems like Valle 2.

Additionally, I value the proposed dataset, but I have reservations about whether the expansion of style tags can be extended to real-world style annotations. This uncertainty arises from the fact that natural language cannot be entirely broken down into a mere combination of tags. To address these concerns, it would be insightful to know if the authors have considered testing the system with free-form style descriptions that go beyond simple tag combinations.

**Questions:**

1. Could you provide further clarification on the rationale behind using different binning thresholds for pitch and speaking rate? Did this lead to significant discrepancies between speaker-level and utterance-level pitch labels, or were other factors at play?

---

### Official Review · Reviewer_XHjY · 2024-10-28

**Soundness:** 2
**Presentation:** 3
**Contribution:** 2
**Rating:** 5
**Confidence:** 4

**Summary:**

The article introduces a crisper definition for style-prompted text-to-speech (TTS) systems, which categorizes the style tags along two dimensions: how the tags are collected (automatic, demographic, and abstract) and what aspects they represent (intrinsic to speaker identity or situational speaking styles). To support this comprehensive tagging definition, the authors annotated a subset of the VoxCeleb dataset with abstract intrinsic style tags, created the StyledVoxCeleb dataset, and utilized existing datasets Expresso and EARS for abstract situational tags. They then fine-tuned the Parler-TTS model using these diverse datasets. Their experiments demonstrate that the new model significantly outperforms baseline systems in maintaining speech-style consistency, achieving higher consistency MOS scores and better tag recall scores.

**Strengths:**

1. The proposed crisper categorization of style tags successfully addresses some limitations of current style-prompted TTS approaches.
2. This work will release its model and collected data upon publication, which would contribute to related research fields.
3. The data collection methodology outlined in Section 3.1 is clear and straightforward. This is also helpful for future research.

**Weaknesses:**

The main weaknesses are:
1. Possibly inappropriate experimental setup. As described in Section 4.1, this paper combines the test splits of StyledVoxCeleb, Expresso, EARS, and LibriTTS-R to construct the dataset for evaluation, which come from different data domains. However, if the test set includes these three datasets, then ``Init.`` -> ``+LTTSR`` -> ``+LTTSP, Exp, EARS`` -> ``Ours`` in Table 5 will surely perform increasingly better as the domain gap gradually diminishes. Therefore, it can not be determined that the performance improvements are due to the proposed crisper categorization of style tags. To support the claims of this paper, the test set should be selected from out-of-domain data.
2. In Table 3, the recall is only 0.36, and the accuracy is just 75%, while similar works have an accuracy of around 85%. Does this suggest that the style description tags in this dataset are inaccurate, resulting in lower testing accuracy?
3. In Table 4, the inclusion of the Exp, EARS, and StyledVoxCeleb datasets led to a noticeable decrease in speech intelligibility. Although the authors hypothesize that this decline is due to the introduction of abstract speaking styles, small dataset sizes, and noisy data, there is a lack of further experiments to support these assumptions.
4. The lack of demo audio examples makes it harder to judge whether the experimental results are convincing. The authors could provide some audio samples.

There are also some minor issues:
1. Clarity issues. In Section 2, Line 139, the phrase ``this does not add a real signal to the dataset`` is unclear. What is the real signal? Are you referring to the variational information [1] in the speech signal? Additionally, in Line 415, there is a typo: ``on read audiobook data`` -> ``on reading-style audiobook data``.
2. In Section 8, the authors claim that the noisy samples from VoxCeleb negatively impact model performance and suggest that scaling to more speakers may mitigate this issue. However, wouldn’t increasing the number of noisy examples further degrade the model’s performance? Have the authors considered using cleaner large-scale datasets, such as Emilia [2]?
3. In Section 3.1, the paper says that ``we can only provide the celebrity’s name rather than the actual speech clip``. Have the authors experimented with any models that support audio input, such as Gemini-pro or Qwen-Audio 2?

To conclude, this paper presents an innovative approach to defining style prompt tags; however, it does not introduce any novel algorithms or model structures, so its contribution is moderately limited. Additionally, as noted in the main weaknesses part, there are still some issues with the experimental setups. Therefore, I give it a score of 5.

[1] Ren, Yi, et al. "Fastspeech 2: Fast and high-quality end-to-end text to speech." arXiv preprint arXiv:2006.04558 (2020).
[2] He, Haorui, et al. "Emilia: An extensive, multilingual, and diverse speech dataset for large-scale speech generation." arXiv preprint arXiv:2407.05361 (2024).
[3] Ji, Shengpeng, et al. "Textrolspeech: A text style control speech corpus with codec language text-to-speech models." ICASSP 2024-2024 IEEE International Conference on Acoustics, Speech and Signal Processing (ICASSP). IEEE, 2024.

**Questions:**

My questions are included in the weaknesses part.

---

### Official Review · Reviewer_zSYK · 2024-11-02

**Soundness:** 2
**Presentation:** 3
**Contribution:** 2
**Rating:** 3
**Confidence:** 5

**Summary:**

The author proposes an improved approach for the style-prompted text-to-speech system capable of generating speech that aligns with specific style prompts. The authors categorize style tags into automatic, demographic, and abstract groups, differentiating between intrinsic tags related to speaker identity (e.g., gender, pitch) and situational tags tied to individual utterances (e.g., emotion). The authors conducted experiments based on the open-source model Parler-TTS, and while the model shows enhanced style consistency, challenges remain in balancing speech quality and content accuracy due to the diversity and quality of the training data. The authors intend to open source their dataset and model to facilitate further research.

**Strengths:**

- The authors present a more detailed definition of style descriptions that was not considered in previous work.
- The authors have greatly expanded the existing space of stylistic tags to cover 63 tags, and this extensive tag coverage opens up the possibility of generating more personalized and diverse speech.
- With more comprehensive style labelling and a diverse dataset, the trained TTS model outperforms the baseline model in terms of speech style consistency metrics.

**Weaknesses:**

- Creating the annotations required significant manual labour, and I acknowledge the effort put in by the authors, but apart from the extended style description definitions, I don't think the article shows the novelty of an ICLR-level paper.
- Both the production of the datasets and the evaluation of the models rely excessively on human subjective opinions. Despite the various measures taken by the authors to ensure quality, it is difficult to avoid introducing inherent biases.
- Despite the improvement in stylistic consistency, the model performs poorly in terms of speech naturalness (MOS) and content accuracy (WER). For TTS models, speech quality and content accuracy are basic requirements, the lack of which will directly affect the practical application value of the dataset.
- The authors present so many categories of labels, but don't analyse them in more detail, and I worry about whether some of the labels will actually make a difference.

**Questions:**

See details in Weaknesses.

---

### Official Review · Reviewer_cevn · 2024-11-04

**Soundness:** 2
**Presentation:** 3
**Contribution:** 2
**Rating:** 5
**Confidence:** 3

**Summary:**

This paper introduces a clearer categorization of style tags for text-to-speech synthesis, expanding to 63 tags across automatic, demographic, and abstract categories. Using datasets like StyledVoxCeleb and the Parler-TTS framework, the model improves on speech-style consistency while facing challenges in quality and content accuracy. The authors plan to open-source their dataset and model to aid further advancements in style-prompted TTS.

**Strengths:**

1. The paper introduces a detailed and clear categorization of speech style tags into abstract intrinsic, abstract situational, demographic, and automatic tags. This clear delineation facilitates a more nuanced understanding and application of style prompts in TTS models.
2. The trained TTS models exhibit improved performance on speech-style consistency metrics compared to baselines, demonstrating the effectiveness of the proposed approach in maintaining consistent speech styles.
3. The commitment to open-source the dataset and models upon publication promotes transparency and further research in the field, providing valuable resources for other researchers and developers.

**Weaknesses:**

Weaknesses:
1. The paper utilizes specific open-source datasets and designs corresponding style tag annotation strategies to enable human annotators to meet annotation requirements, thus expanding the range of abstract style tag data. This approach is relatively common in prior work, such as VoxEditor \[1] although it does not cover all types comprehensively.
2. The current cost of manual annotation limits the scale of building a style tag dataset, which is undoubtedly insufficient for training an effective style-prompted TTS or even a zero-shot style-prompted TTS. This issue is evidenced in the experiments, where the speech MOS scores and content consistency metrics are inferior to those of the baseline models. Additionally, some tags leverage existing metadata from open-source datasets, such as Expresso and EARS, which are rich in emotional tags, but this restricts the expansion of dataset size due to the limited scale of available open-source emotional datasets. Expanding to larger datasets and more speakers is a crucial issue that needs resolution.
3. The experiments related to the style-prompted TTS, including comparisons with baselines, should include some demo audio examples to provide a more intuitive comparison. Additionally, for the constructed text style dataset, providing some sample audio style prompts to demonstrate the annotation effect would be beneficial.

\[1]:  Sheng Z, Ai Y, Liu L J, et al. Voice Attribute Editing with Text Prompt[J]. arXiv preprint arXiv:2404.08857, 2024.

**Questions:**

See details in Paper Weaknesses.

Another issue is that this paper introduces a wide variety of style tags, which might lead to potential conflicts between tags, such as Rhythm and Speaking Rate Levels, or Pitch Levels and Emotion. For example, the emotion of Sleepy typically involves low pitch. If a style text prompt includes both High-pitched and Sleepy emotions and is input into the model, how would it perform?

---

### Note · Authors · 2024-11-15

**Comment:**

We are actively working on revising the paper to improve scalability of our data collection approach, train on cleaner data to prevent degradation of TTS metrics, and perform more analyses on what abstract tags matter. However, this will result in a larger revision than possible during rebuttal, and hence we withdraw the submission.

**Withdrawal Confirmation:**

I have read and agree with the venue's withdrawal policy on behalf of myself and my co-authors.